# Trajectory Flow Matching with Applications to Clinical Time Series Modeling

**Xi Zhang**[1,2] *     **Yuan Pu**[3] *     **Yuki Kawamura**[4]     **Andrew Loza**[3]
**Yoshua Bengio**[2,5,6]     **Dennis L. Shung**[3] †     **Alexander Tong**[2,5] †

[1]McGill University, [2]Mila - Quebec AI Institute,
[3]Yale School of Medicine
[4]School of Clinical Medicine, University of Cambridge,
[5]Université de Montréal, [6]CIFAR Fellow

## Abstract

Modeling stochastic and irregularly sampled time series is a challenging problem found in a wide range of applications, especially in medicine. Neural stochastic differential equations (Neural SDEs) are an attractive modeling technique for this problem, which parameterize the drift and diffusion terms of an SDE with neural networks. However, current algorithms for training Neural SDEs require backpropagation through the SDE dynamics, greatly limiting their scalability and stability. To address this, we propose **Trajectory Flow Matching** (TFM), which trains a Neural SDE in a *simulation-free* manner, bypassing backpropagation through the dynamics. TFM leverages the flow matching technique from generative modeling to model time series. In this work we first establish necessary conditions for TFM to learn time series data. Next, we present a reparameterization trick which improves training stability. Finally, we adapt TFM to the clinical time series setting, demonstrating improved performance on four clinical time series datasets both in terms of absolute performance and uncertainty prediction, a crucial parameter in this setting.

## 1 Introduction

Real world problems often involve systems that evolve continuously over time, yet these systems are usually noisy and irregularly sampled. In addition, real-world time series often relate to other covariates, leading to complex patterns such as intersecting trajectories. For instance, in the context of clinical trajectories in healthcare, patients' vital sign evolution can follow drastically different, crossing paths even if the initial measurements are similar, due to the influence of the covariates such as medication intervention and underlying health conditions. These covariates can be time-varying or static, and often sparse.

Differential equation-based dynamical models are proficient at learning continuous variables without imputations [Chen et al., 2018, Rubanova et al., 2019, Kidger et al., 2021b]. Nevertheless, systems governed by ordinary differential equations (ODEs) or stochastic differential equations (SDEs) are unable to accommodate intersecting trajectories, and thus requires modifications such as augmentation or modelling higher-order derivatives [Dupont et al., 2019]. While ODEs model deterministic systems, SDEs contain a diffusion term and can better represent the inherent uncertainty and fluctuations present in many real world systems. However, fitting stochastic equations to real life

---

*Joint first authorship

†Joint senior authorship. Correspondence to `alexander.tong@mila.quebec`
   Code available at: `https://github.com/nZhangx/TrajectoryFlowMatching`

38th Conference on Neural Information Processing Systems (NeurIPS 2024).

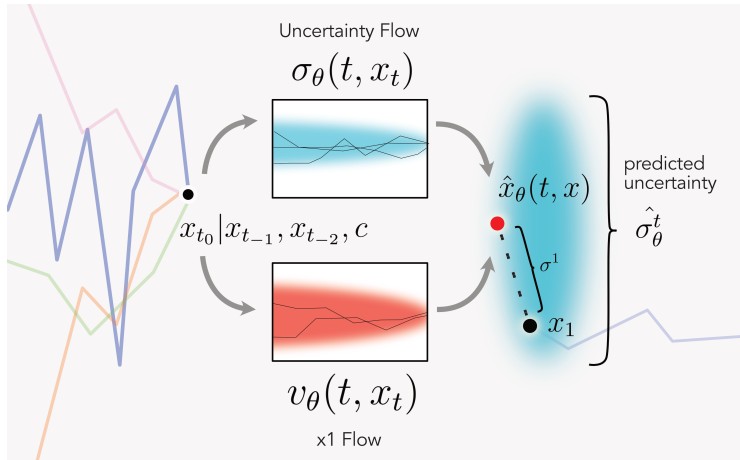

Figure 1: Trajectory Flow Matching trains both an estimator of the next timepoint ($\hat{x}_\theta(t, x)$) and an estimation of the uncertainty ($\sigma_\theta(t, x_t)$). Using the conditional flow matching framework, these can be used to predict the instantaneous velocity $v_\theta(t, x_t)$ and future observations. Both flows are conditioned on past data $x_{[t-h,t-1]}$ and conditional variables $c$.

data is challenging because they have thus far required time-consuming backpropagation through an SDE integration.

In the domain of generative models, diffusion models [Ho et al., 2020, Nichol and Dhariwal, 2021, Song et al., 2021] and more recently flow matching models [Lipman et al., 2023, Albergo et al., 2023, Li et al., 2020] have had enormous success by training dynamical models in a *simulation-free* framework. The simulation-free framework facilitates the training of much larger models with significantly improved speed and stability. In this work we generalize simulation-free training for fitting stochastic differential equations to time-series data, to learn population trajectories while preserving individual characteristics with conditionals. We present this method as **Trajectory Flow Matching**. We demonstrate that our method outperforms current state of the art time series modelling architecture including RNN, ODE based and flow matching methods. We empirically demonstrate the utility of our method in clinical applications where hemodynamic trajectories are critical for ongoing dynamic monitoring and care. We applied our method to the following longitudinal electronic health record datasets: medical intensive care unit (MICU) data of patients with sepsis, ICU patients at risk for cardiac arrest, Emergency Department (ED) data of patients with acute gastrointestinal bleeding, and MICU data of patients with acute gastrointestinal bleeding.

Our main contributions are:

- We prove the conditions under which continuous time dynamics can be trained simulation-free using matching techniques.
- We extend the approach to irregularly sampled trajectories with a *time predictive loss* and to estimate uncertainty using an *uncertainty prediction loss*.
- We empirically demonstrate that our approach reduces the error by 15-83% when applied to the real world clinical data modelling.

## 2  Preliminaries

### 2.1  Notation

We consider the setting of a distribution of trajectories over $\mathbb{R}^d$ denoted $\mathcal{X} := \{x^1, x^2, \ldots, x^n\}$ where each $x^i$ is a vector of $T$ datapoints i.e. $x^i := \{x_1^i, x_2^i, \ldots, x_T^i\}$ with associated times $t^i := \{t_1^i, t_2^i, \ldots, t_T^i\}$. Let $x_{[t-h,t-1]}^i$ denote a vector of the last $h$ observed time points. We denote a (Lipschitz smooth) time dependent vector field conditioned on arbitrary conditions $c \in \mathbb{R}^e$ $v(t, x_t, x_{[t-h,t-1]}, c) \rightarrow \frac{dx}{dt} : ([0,1], \mathbb{R}^d, \mathbb{R}^{h \times d}, \mathbb{R}^e) \rightarrow \mathbb{R}^d$ with flow $\phi_t(v)$ which induces the time-dependent density $p_t = \phi_t(v)_\#(p_0)$ for any density $p_0 : \mathbb{R}^d \rightarrow \mathbb{R}_+$ with $\int_{\mathbb{R}^d} p_0 = 1$. We also consider the coupling $\pi(x_0, x_1)$ which operates on the product space of marginal distributions $p_0, p_1$.

## 2.2 Neural Stochastic Differential Equations

A stochastic differential equation (SDE) can be expressed in terms of a smooth drift $f : [0, T] \times \mathbb{R}^d \to \mathbb{R}^d$ and diffusion $g : [0, T] \times \mathbb{R}^d \to \mathbb{R}^{d^2}$ in the Ito sense as:

$$dx_t = f \, dt + g \, dW_t$$

where $W_t : [0, T] \to \mathbb{R}^d$ is the $d$-dimensional Wiener process. A density $p_0(x_0)$ evolved according to an SDE induces a collection of marginal distributions $p_t(x_t)$ viewed as a function $p : [0, T] \times \mathbb{R}^d \to \mathbb{R}_+$. In a *Neural* SDE [Li et al., 2020, Kidger et al., 2021a,b] the drift and diffusion terms are parameterized with neural networks $f_\theta(t, x_t)$ and $g_\theta(t, x_t)$.

$$dx_t = f_\theta(t, x_t)dt + g_\theta(t, x_t)dW_t \tag{1}$$

where the goal is to select $\theta$ to enforce $x_T \sim X_{true}$ for some distributional notion of similarity such as the Wasserstein distance [Kidger et al., 2021b] or Kullback-Leibler divergence [Li et al., 2020]. However, these objectives are *simulation-based*, requiring a backpropagation through an SDE solver, which suffers from severe speed and stability issues. While some issues such as memory and numerical truncation can be ameliorated using the adjoint state method and advanced numerical solvers [Kidger et al., 2021b], optimization of Neural SDEs is still a significant issue.

We note that in the special case of zero-diffusion (i.e. $g_\theta(t, x_t) = 0$) this reduces to a neural *ordinary* differential equation (Neural ODE) [Chen et al., 2018], which is easier to optimize than SDEs, but still presents challenges to scalability.

## 2.3 Matching algorithms

Matching algorithms are a *simulation-free* class of training algorithms which are able to bypass backpropagation through the solver during training by constructing the marginal distribution as a mixture of tractable conditional probability paths.

The marginal density $p_t$ induced by eq. 1 evolves according to the *Fokker-Plank* equation (FPE):

$$\partial_t p_t = -\nabla \cdot (p_t f_t) + \frac{g^2}{2} \Delta p_t \tag{2}$$

where $\Delta p_t = \nabla \cdot (\nabla p_t)$ denotes the *Laplacian* of $p_t$ and gradients are taken with respect to $x_t$.

Matching algorithms first construct a factorization of $p_t$ into conditional densities $p_t(x_t|z)$ such that $p_t = \mathbb{E}_{q(z)}[p_t(x_t|z)]$ and where $p_t(x_t|z)$ is generated by an SDE $dx_t = v_t(x_t|z)dt + \sigma_t(x_t|z)dW_t$. Given this construction it can be shown that the minimizer of

$$\mathcal{L}_{\text{match}}(\theta) := \mathbb{E}_{t,q(z),p_t(x|z)} \left[ \|f_\theta(t, x_t) - v_t(x_t|z)\|^2 + \lambda_t^2 \|g_\theta(t, x_t) - \sigma_t(x_t|z)\|^2 \right] \tag{3}$$

satisfies the FPE of the marginal $p_t$. This is especially useful in the generative modeling setting where $q_0$ is samplable noise (e.g. $\mathcal{N}(0, 1)$) and $q_1$ is the data distribution. Then we can define $z := (x_0, x_1)$ as a tuple of noise and data with $q(z) := q_0(x_0) \otimes q_1(x_1)$. This makes eq. 3 optimize a model which will draw new samples according to the data distribution $q_1(x_1)$ using

$$x_0 \sim q_0; \quad x_1 = \int_0^1 f_\theta(t, x_t)dt + g_\theta(t, x_t)dW_t \tag{4}$$

with the integration computed numerically using any off-the-shelf SDE solver. While this is guaranteed to preserve the distribution over time, it is not guaranteed to preserve the *coupling* of $q_0$ and $q_1$ (if given).

**Paired bridge matching** In generative modeling random pairings [Liu et al., 2023c, Albergo and Vanden-Eijnden, 2023, Albergo et al., 2023] or optimal transport [Tong et al., 2024, Pooladian et al., 2023] pairings are constructed for the conditional distribution $q(z)$. However, in some problems we would like to match pairs of points as is the case in image-to-image translation [Isola et al., 2017, Liu et al., 2023a, Somnath et al., 2023]. In this case, training data comes as pairs $(x_0, x_1)$. In this case we set $q(z) := q(x_0, x_1)$ to be samples from these known pairs, and optimize eq. 3. While empirically, these models perform well, there are no guarantees that the coupling will be preserved outside of the special case when data comes from the (entropic) optimal transport coupling $\pi_\varepsilon^*(q_0, q_1)$ and defined as:

$$\pi_\varepsilon^*(q_0, q_1) = \underset{\pi \in U(q_0, q_1)}{\arg \min} \int d(x_0, x_1)^2 \, d\pi(x_0, x_1) + \varepsilon \, \text{KL}(\pi \| q_0 \otimes q_1), \tag{5}$$

**Algorithm 1** General Trajectory Flow Matching

---

**Input:** Trajectories $\mathcal{X} = \{x^1, x^2, \ldots, x^n\}$, noise $\sigma$, initial networks $v_\theta$ and $\sigma_\theta$.
**while** Training **do**

$\quad x^i \sim \mathcal{U}(\mathcal{X}), \quad k \sim \mathcal{U}\{1, T-1\}, \quad t \sim \mathcal{U}(0,1)$
$\quad \mu_t \leftarrow (1-t)x_k^i + tx_{k+1}^i$
$\quad x_t \sim \mathcal{N}(\mu_t, \sigma^2 t(1-t)I)$
$\quad \mathcal{L}_{\text{TFM}}(\theta) \leftarrow \left\| v_\theta(k+t, x_t) - \frac{x_{k+1}^i - x_t}{1-t} \right\|^2$
$\quad \mathcal{L}_{\sigma_t}(\theta) \leftarrow \left\| \sigma_\theta(k+t, x_t) - \mathcal{L}_{\text{TFM}} \right\|^2$
$\quad \theta \leftarrow \text{Update}(\theta, \nabla_\theta \mathcal{L}_{\text{TFM}}(\theta), \nabla_\theta \mathcal{L}_{\sigma_t}(\theta))$
**return** $v_\theta, \sigma_\theta$

---

where $U(q_0, q_1)$ is the set of admissible transport plans (i.e. joint distributions over $x_0$ and $x_1$ whose marginals are equal to $q_0$ and $q_1$) as shown in [Shi et al., 2023] for some regularization parameter $\varepsilon \in \mathbb{R}_{\geq 0}$.

## 3 Trajectory Flow Matching

We now describe our simulation-free method to learn SDEs on time-series data using *trajectory flow matching* as summarized in Alg. 1. In the case of time series we need to ensure that trajectory couplings are preserved. We first set out a general algorithm for flow matching on vector fields in §3.1 then present a numerical reparameterization which we find stabilizes training in §3.2, a next observation prediction for irregularly sampled time series in §3.3, and finally present how to learn the noise in §3.4.

### 3.1 Preserving Couplings

In this section, we assume access to fully observed and evenly spaced trajectories $\mathcal{X} = (x^1, x^2, \ldots, x^n)$ with $x^i := (x_1^i, x_2^i, \ldots, x_T^i)$ for clarity and notational simplicty. We note that our method is easily extensible to the more general setting of irregularly sampled trajectories. In this simplified case we let

$$z := (x_1, x_2, \ldots, x_T) \tag{6}$$

$$q(z) := \mathcal{U}(\mathcal{X}) \tag{7}$$

$$p_t(x|z) := \mathcal{N}((\lceil t \rceil - t)x_{\lfloor t \rfloor} + (t - \lfloor t \rfloor)x_{\lceil t \rceil}, \sigma^2(\lceil t \rceil - t)(t - \lfloor t \rfloor)\mathbf{I}) \tag{8}$$

$$u_t(x|z) := \frac{x_{\lceil t \rceil} - x_t}{\lceil t \rceil - t} \tag{9}$$

where $\mathcal{U}(\mathcal{X})$ is the uniform empirical distribution over $\mathcal{X}$, $\lceil \cdot \rceil$, $\lfloor \cdot \rfloor$ are the ceiling and floor functions, and $\mathcal{N}(\cdot, \cdot)$ is the multivariate normal distribution. This is a valid regression in the sense that a function minimized with Alg. 1 will return a stochastic process that will match the observed marginal distributions over time as shown in the following lemma.

**Lemma 3.1.** *The SDE $dx_t = u_t(x|z)dt + \sigma^2 dW_t$ where $u_t$ is defined in eq. 9 generates $p_t(x|z)$ in eq. 8 with initial condition $p_0 := \delta_{x_1}$ where $\delta$ is the Dirac delta function.*

however, while useful, this is still insufficient for time series modeling, as it does not ensure coupling preservation. For intuition why this is an issue see Figure 2.

In TFM we ensure that the couplings are preserved for history lengths $h > 0$. i.e. $\hat{\pi}(x_{T-h}, x_{T-h+1}, \ldots, x_T) = \pi(x_{T-h}, x_{T-h+1}, \ldots, x_T)$. We first establish a method to ensure that these couplings are preserved allowing us to use simulation-free flow matching training for the time-series modeling task. Specifically, as long as the model takes as input $(x_{T-h}, x_{T-h+1}, \ldots, x_T)$ in predicting the flow from $T \to T+1$, then there exists a function $f_\theta(X_{T-h:T})$ such that the coupling is preserved.

**Proposition 3.2** (Coupling Preservation). *Under mild regulatory criteria on $u_t(\cdot|z)$, $p_t$, and $q$, if*

$$\mathbb{E}_{t \sim \mathcal{U}(0,T), z \sim q(z), c \sim q(c|z), x_t \sim p_t(x_t|z)} \| u_t(x_t|z, c) - u_t(x_t|c) \|_2^2 = 0$$

*and $z, q(z), p_t(x|z)$ and $u_t(x|z)$ are as defined in eqs. 6-9 then $\Pi(u)^\star = \Pi^\star(x_{1:T})$.*

Where $\Pi(u)^\star$ represents the coupling of a model which attains minimal loss according to eq. 3 and $\Pi^\star(x_{1:T})$ is the coupling of the data distribution. Intuitively, as long as no two paths cross given conditionals $c$, then the coupling is preserved. In prior work $c = \emptyset$, and the coupling is only preserved in special cases such as eq. 5.

We next enumerate three assumptions under which the coupling is guaranteed to be preserved at the optima. We note that these are

**(A1)** When $c = x_0$ and there exists $T : \mathcal{X} \to \mathcal{X}$ such that $T(x_0) = x_1$ iff $\Pi^\star(x_0, x_1)$. We note that this is equivalent to asserting the existence of a Monge map $T^\star$ for the coupling $\Pi^\star$.

**(A2)** There exist no two trajectories $x^i$, $x^j$ such that $x_t^i = x_t^j$ for $h$ consecutive observations and $g = 0$.

**(A3)** Trajectories are associated with unique conditional vectors $c$ independent of $t$.

Even in cases when **(A1)-(A3)** may not hold exactly, TFM is a useful model and can often still learn useful models of the data. In some sense uniqueness up to some history length is enough as it shows TFM is as powerful as discrete-time autoregressive models. Proofs and further examples are available in §A.1.

### 3.2 Target prediction reparameterization

While flow matching generally predicts the flow, there is a target predicting equivalent namely given $v_\theta(t, x) := \frac{\hat{x}_\theta^{\lceil t \rceil}(t, x_t) - x_t}{\lceil t \rceil - t}$ and $u_t(x|z) := \frac{x^{\lceil t \rceil} - x_t}{\lceil t \rceil - t}$ which is equivalent to $x_1 - x_0$ when $x_t : tx_1 + (1-t)x_0$ then it is easy to show that the target predicting loss is equivalent to a time-weighted flow-matching loss. Specifically let the target predicting loss be

$$\mathcal{L}_{\text{target}}(\theta) = \mathbb{E}_{t,q(z),p_t(x|z)} \|\hat{x}_\theta^{\lceil t \rceil}(t, x) - x^{\lceil t \rceil}\|^2 \tag{10}$$

then it is easy to show that

**Proposition 3.3.** *There exists a scaling function $c(t) : \mathbb{R}_+ \to \mathbb{R}$ such that $\mathcal{L}_{target}(\theta) = c(t)\mathcal{L}_{match}(\theta)$.*

### 3.3 Irregularly sampled trajectories

We next consider irregularly sampled time series of the form $x^i := \left((x_1^i, t_1^i), (x_2^i, t_2^i), \ldots, (x_T^i, t_T^i)\right)$ with $t_1^i < t_2^i < \cdots < t_T^i$ with $t_{next}$ denoting the next timepoint observed after time $t$. In this case, when combined with the target predicting reparameterization in §3.2, we can predict the time till next observation. We therefore parameterize an auxiliary model $h_\theta(t, x_t) : [0, T] \times \mathbb{R}^d \to [0, T]$ which predicts the next observation time. This is useful numerically, but also, perhaps more importantly, is useful in a clinical setting, where the spacing between measurements can be as informative as the measurements themselves [Allam et al., 2021]. $h_\theta$ is trained to predict the time till the next observation with the *time predictive loss*:

$$\mathcal{L}_{\text{tp}}(\theta) = \sum_{t \in \mathcal{T}^i} \|h_\theta(t, x_t) - (t_{\text{next}} - t)\|_2^2 \tag{11}$$

where $t_{\text{next}}$ is the time of the next measurement. This can be used in conjunction with the $x_{\text{next}}$ predictor to calculate the flow at time $t$ as

$$v_\theta(t, x_t) := \frac{\hat{x}_\theta^1(t, x_t) - x_t}{h_\theta(t, x_t) - t} \tag{12}$$

which can be used for inference on new trajectories.

### 3.4 Uncertainty prediction

Finally, we consider uncertainty prediction. till now we have defined conditional probability paths using a fixed noise parameter $\sigma$. However, this does not have to be fixed. Instead, we consider a *learned* $\sigma_\theta(t, x_t)$ which can be learned iteratively with the loss:

$$\mathcal{L}_{\text{uncertainty}}(\theta, x) = \sum_{t \in \mathcal{T}} \left\| \sigma_\theta(t, x_t) - \|\hat{x}_\theta(t, x_t) - x_{\text{next}}\|_2^2 \right\|_2^2 \tag{13}$$

which learns to predict the error in the estimate of $x_t$. This loss can be interpreted as training an epistemic uncertainty predictor which is similar to that proposed in direct epistemic uncertainty prediction (DEUP) [Lahlou et al., 2023].

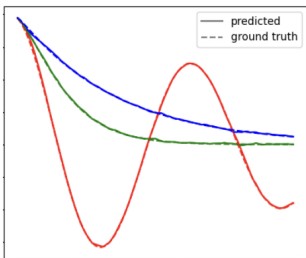 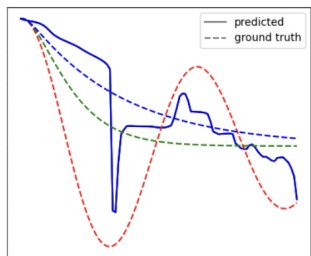 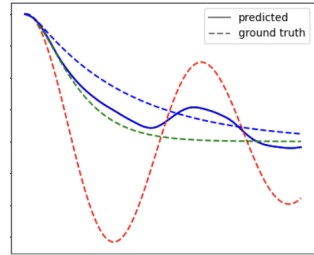

Figure 2: 1D harmonic oscillator overfitting experiment results. **Left:** TFM-ODE (ours) with memory = 3. **Middle:** TFM-ODE (ours) without memory. **Right:** Aligned FM [Liu et al., 2023a, Somnath et al., 2023].

## 4  Experimental Results

In this section we empirically evaluate the performance of the trajectory flow matching objective in terms of time series modeling error, but also uncertainty quantification. We also evaluate a variety of simulation-based and simulation-free methods including both stochastic and deterministic methods. Stochastic methods are in general more difficult to fit, but can be used to better model uncertainty and variance. Further experimental details can be found in §B. Experiments were run on a computing cluster with a heterogenous cluster of NVIDIA RTX8000, V100, A40, and A100 GPUs for approximately 24,000 GPU hours. Individual training runs require approximately one gpu day.

**Baselines**  In addition to different ablations of trajectory flow matching, we also evaluate NeuralODE [Chen et al., 2018], NeuralSDE [Li et al., 2020, Kidger et al., 2021b, Kidger, 2022], Latent NeuralODE [Rubanova et al., 2019], and an aligned flow matching method (Aligned FM) [Liu et al., 2023a, Somnath et al., 2023] where the couplings are sampled according to the ground truth coupling during training.

**Metrics**  We primarily make use of two metrics. The average mean-squared-error (Mean MSE) over left out time series to measure the time series modeling error defined as

$$\text{MSE}(\hat{x}, x) = \frac{1}{T-1} \sum_{t \in [2,T]} \|\hat{x}_t - x_t\|_2^2, \tag{14}$$

where $\hat{x}$ and $x$ are the predicted and true trajectories respectively. We also use the *maximum mean discrepancy* with a radial basis function kernel (RBF MMD) which measures how well the distribution over next observation is modelled by comparing the predicted distribution to the distribution over next states in the ground truth trajectory. Specifically we compute:

$$\text{RBF-MMD}(\theta, \hat{x}, x) := \frac{1}{T-1} \sum_{t \in [2,T]} \text{MMD}(\hat{\Delta}_t, \Delta_t) \tag{15}$$

where $\hat{\Delta}_t = \hat{x}_t - x_{t-1}$, $\Delta_t = x_t - x_{t-1}$, and $\hat{x}_t := \int_{s=t-1}^{t} f_\theta(s, x_s)ds + g_\theta(t, x_s)dW_s$ is a set of samples from the model prediction at time $t$.

### 4.1  Exploring coupling preservation with 1D harmonic oscillators

We begin by evaluating how trajectory flow matching performs in a simple one dimensional setting of harmonic oscillators. We show that the canonical conditional flow and bridge matching [Liu et al., 2023c,b, Albergo and Vanden-Eijnden, 2023], specifically aligned approaches [Somnath et al., 2023, Liu et al., 2023a] are unable to preserve the coupling even in a simple one dimensional setting. However, augmented with our trajectory flow matching approach, and specifically using **(A2)**, which includes information on previous observations, the model is able to fit the harmonic oscillator dataset well.

The harmonic oscillator dataset consists of one-dimensional oscillatory trajectories from a damped harmonic oscillator, with each trajectory distinguished by a unique damping coefficient $c$. Specifically we sample trajectories $x$ from:

$$x_i = x_{i-1} + v_{i-1}(t_i - t_{i-1}); \quad x_0 = 1 \tag{16}$$

where $v$ is the velocity of the oscillator updated by

$$v_i = v_{i-1} + \left( -\frac{c}{m} v_{i-1} - \frac{k}{m} x_{i-1} \right) (t_i - t_{i-1}); \quad v_0 = 0 \tag{17}$$

with $t_i = 0.1 \cdot i$ for $i = 0, 1, 2, \ldots, 99$, spring constant $k = 1$, and mass $m = 1$.

As $c$ increases, the trajectories evolve from underdamped scenarios with prolonged oscillations to critically and overdamped states where the oscillator quickly stabilizes. This leads to intersecting trajectories due to frequency and phase differences, despite their shared starting point. We perform overfitting experiments on three trajectories generated by varying $c$.

As shown in Figure 2, models without history information are unable to distinguish between the three crossing trajectories that share the same starting point, resulting in overlapping predictions. In contrast, TFM-ODE that incorporates three previous observations is able to fit the crossing trajectories with high accuracy, with the predicted trajectories almost completely overlapping the ground truth. This is because the dataset with satisfies **(A2)** with $h = 4$ (TFM-ODE), but not $h = 0$ (TFM-ODE no memory and Aligned FM).

## 4.2 Experiments on clinical datasets

Next we compared the performance of TFM and TFM-ODE with the current SDE and ODE baselines, respectively, for modeling real-world patient trajectories formed with heart rate and mean arterial blood pressure measurements within the first 24 hours of admission across four different datasets. These are clinical measurements that are taken most frequently and used to evaluate the hemodynamic status of patients, a key indicator of disease severity. Additionally, we evaluated our models against flow matching on these datasets, each with distinct characteristics, to assess their ability to generalize across different distributions. A full description of the datasets are available in Appendix B.2 with the publicly available datasets used under The PhysioNet Credentialed Health Data License Version 1.5.0 and the EHR dataset with local institutional IRB approval:

- **ICU Sepsis:** a subset of the eICU Collaborative Research Database v2.0 [Pollard et al., 2019] of patients admitted with sepsis as the primary diagnosis.
- **ICU Cardiac Arrest:** a subset of the eICU Collaborative Research Database v2.0 [Pollard et al., 2019] of patients at risk for cardiac arrest.
- **ICU GIB:** a subset of the Medical Information Mart for Intensive Care III [Johnson et al., 2016] of patients with gastrointestinal bleeding as the primary diagnosis.
- **ED GIB:** patients presenting with signs and symptoms of acute gastrointestinal bleeding to the emergency department of a large tertiary care academic health system.

### 4.2.1 Prediction accuracy and precision: TFM and TFM-ODE

**TFM-ODE yields more accurate trajectory prediction**    Across the four datasets TFM-ODE outperformed the baseline models by $15\%$ to $20\%$, as seen in table 1. We noticed that TFM has a similar performance as TFM-ODE. In one case TFM outperformed the non-stochastic TFM-ODE, as seen in the ICU GIB dataset. For ICU sepsis, the performance improvement from the baseline is the most significant, around $83\%$. This coincides with the ICU sepsis dataset having the most amount of measurement per trajectory. The improvement is seen in both TFM and TFM-ODE, possibly indicating they are able to learn better given more data, resulting in a more precise flow. Not formally measured, we noted that given the same time constraint, FM based models were significantly faster and often finished training before the time limit.

**TFM yields better uncertainty prediction**    Though TFM-ODE had lower test MSE for half of the times, TFM yielded better uncertainty prediction overall, as seen in table 2. Notably, TFM also had less variance in the uncertainty prediction than TFM-ODE. A plausible explanation in this case is a sacrifice in bias that subsequently decreases the variance for the stochastic implementation, reflecting the bias-variance trade off. Sampled graphs of TFM can be seen in figure 3. It is notable that the model is able to detect the measurement uncertainty at certain timepoints, matching the increase in amplitude of oscillation in patient trajectories.

### 4.2.2 Trajectory Variance Distribution Comparison

**TFM trajectories accurately match the noise distribution in the data**    TFM is able to match the noise distribution in addition to the overall trajectory shape, which is useful in settings where

Table 1: Mean $\pm$ Std. deviation MSE ($\times 10^{-3}$) by models and datasets. Split into deterministic (top) and stochastic models (bottom). Top performing model for each setting and dataset in **bold**.

| | ICU Sepsis | ICU Cardiac Arrest | ICU GIB | ED GIB |
|---|---|---|---|---|
| NeuralODE | $4.776 \pm 0.000$ | $6.153 \pm 0.000$ | $3.170 \pm 0.000$ | $10.859 \pm 0.000$ |
| FM baseline ODE | $4.671 \pm 0.791$ | $10.207 \pm 1.076$ | $118.439 \pm 17.947$ | $11.923 \pm 1.123$ |
| LatentODE RNN | $61.806 \pm 46.573$ | $386.190 \pm 558.140$ | $422.886 \pm 431.954$ | $980.228 \pm 1032.393$ |
| TFM-ODE (ours) | $\mathbf{0.793 \pm 0.017}$ | $2.762 \pm 0.021$ | $2.673 \pm 0.069$ | $\mathbf{8.245 \pm 0.495}$ |
| NeuralSDE | $4.747 \pm 0.000$ | $3.250 \pm 0.024$ | $3.186 \pm 0.000$ | $10.850 \pm 0.043$ |
| TFM (ours) | $0.796 \pm 0.026$ | $\mathbf{2.755 \pm 0.015}$ | $\mathbf{2.596 \pm 0.079}$ | $8.613 \pm 0.260$ |

Table 2: Uncertainty test MSE loss for TFM-ODE and TFM with two different ICU datasets.

| | ICU sepsis | ICU Cardiac Arrest | ICU GIB |
|---|---|---|---|
| TFM-ODE | $1.039 \pm 0.1645$ | $0.970 \pm 0.1426$ | $0.9843 \pm 0.2233$ |
| TFM | $\mathbf{0.724 \pm 0.0072}$ | $\mathbf{0.636 \pm 0.0024}$ | $\mathbf{0.605 \pm 0.0137}$ |

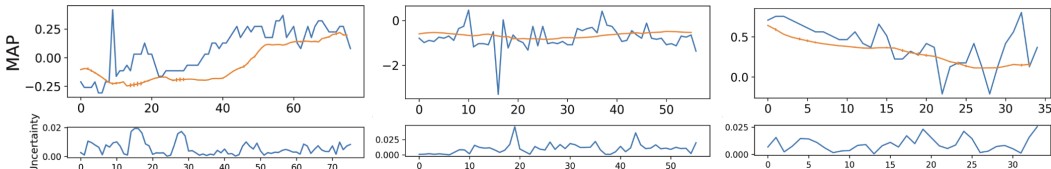

Figure 3: Three samples from predicted trajectory and uncertainty on ICU GIB test set. **Top**: Predicted (orange) and the ground truth (blue) mean arterial pressure (MAP). **Bottom**: The absolute value of the uncertainty predicted by TFM.

data has high stochasticity. We compared our models to NeuralODE and NeuralSDE in matching the variance in neighboring data points, seen in table 3. We verify that between the baseline NeuralSDE and NeuralODE, NeuralSDE has a lower MMD and is better able to match data points. We find in ICU GIB and ED GIB datasets, TFM outperforms both in matching the variance in data. Notably, the performance pattern is reversed for the MMD metrics and mean MSE metrics with respect to TFM and TFM-ODE where better MSE leads to worse MMD and vice versa. As such, this further confirms the bias-variance trade-off for both TFM and TFM-ODE implementation.

Table 3: Data variance MMD for by models and datasets. Split into deterministic models (top) and stochastic models (bottom). Top performing model for each setting and dataset in **bold**.

| | ICU Sepsis | ICU Cardiac Arrest | ICU GIB | ED GIB |
|---|---|---|---|---|
| NeuralODE | $1.988 \pm 0.000$ | $2.246 \pm 0.000$ | $2.090 \pm 0.000$ | $2.192 \pm 0.000$ |
| TFM-ODE (ours) | $\mathbf{1.172 \pm 0.017}$ | $1.295 \pm 0.006$ | $1.087 \pm 0.02$ | $1.063 \pm 0.031$ |
| NeuralSDE | $1.212 \pm 0.000$ | $3.261 \pm 0.020$ | $1.332 \pm 0.000$ | $1.465 \pm 0.122$ |
| TFM (ours) | $1.199 \pm 0.006$ | $\mathbf{0.993 \pm 0.003}$ | $\mathbf{0.844 \pm 0.013}$ | $\mathbf{0.717 \pm 0.016}$ |

### 4.2.3 Ablation Study

We performed ablation studies on TFM and TFM-ODE to attribute importance of various model components contributing to the performance, as seen in table 4. We examined three aspects of the model, two of which were part of our main contributions: uncertainty prediction and memory. We also ablate the model hidden dimension width to infer its potential in scaling effect.

**TFM and TFM-ODE performance scales with model size** In contrast to Neural DE based models, TFM and TFM-ODE exhibit scaling effect, in which the model performance becomes better with a larger hidden dimension. This has been observed in previous flow matching models in image generation [Tong et al., 2024]. This may pave the way for further improvements from larger models.

Table 4: Mean MSE ($\times 10^{-3}$) by ablated versions of TFM, TFM-ODE, and datasets.

| | Uncertainty Prediction | Memory | Hidden Size | ICU Sepsis | ICU Cardiac Arrest | ICU GIB | ED GIB |
|---|---|---|---|---|---|---|---|
| TFM-ODE | ✓ | ✓ | 256 | **0.793 ± 0.017** | 2.762 ± 0.017 | 2.673 ± 0.069 | 8.245 ± 0.495 |
| | | ✓ | 256 | 1.170 ± 0.014 | 2.759 ± 0.015 | 3.097 ± 0.054 | 8.659 ± 0.429 |
| | | | 256 | 1.555 ± 0.122 | 3.242 ± 0.050 | 2.981 ± 0.161 | **6.381 ± 0.451** |
| | | | 64 | 1.936 ± 0.262 | 3.244± 0.025 | 4.003 ± 0.347 | 11.253± 4.597 |
| TFM | ✓ | ✓ | 256 | 0.796 ± 0.026 | **2.596 ± 0.079** | **2.762 ± 0.021** | 8.613 ± 0.260 |
| | | ✓ | 256 | 0.816 ± 0.031 | 2.778 ± 0.021 | 2.754 ± 0.095 | 8.600 ± 0.389 |
| | | | 64 | 1.965 ± 0.289 | 3.271 ± 0.031 | 4.037 ± 0.314 | 7.549 ± 0.737 |

**Uncertainty improves performance of trajectory prediction**  For TFM and TFM-ODE, the flow network used to learn the uncertainty $\sigma_{x_t}$ is separate from the flow network learning $x_t$. The loss function of the network learning $x_t$ is independent of uncertainty flow network. Therefore, it was unexpected that taking away the uncertainty prediction would result in increased MSE test loss for learning $x_t$. This implies further a process in the synergistic effects between $x_t$ flow and $\sigma_{x_t}$ flow.

**Trajectory memory may improve performance in high frequency measurement settings**  We conditioned the model based on a sliding window of trajectory history to disentangle data points that otherwise look indistinguishable to FM models. This improved the interpolation performance in the ICU Sepsis and ICU GIB dataset. Notably, this modification did not improve performance the ED GIB dataset, which could be due to shorter trajectories for patients and lower measurement frequency in the defined time period. This may also be explained by the decreased severity of disease in the ED compared to the ICU. Adding memory as a condition may be more suitable for patients whose clinical trajectories have a higher frequency of measurements.

## 5   Related Work

Continuous-time neural network architectures have outperformed traditional RNN methods in modeling irregularly sampled clinical time series to optimize interpolation and extrapolation. Neural ODE with latent representations of trajectories [Rubanova et al., 2019] outperformed RNN-based approaches [Lipton et al., 2016, Che et al., 2018, Cao et al., 2018, Rajkomar et al., 2018] for interpolation while providing explicit uncertainty estimates about latent states. More recently, Neural SDEs appear to outperform LSTM [Hochreiter and Schmidhuber, 1997], Neural ODE [Chen et al., 2018, De Brouwer et al., 2019, Dupont et al., 2019, Lechner and Hasani, 2020], and attention-based [Shukla and Marlin, 2021, Lee et al., 2022] approaches in interpolation performance while natively handling uncertainty using drift and diffusion terms [Oh et al., 2024].

Discrete-time approaches offer an alternative to our continuous-time model model transformers utilize a discrete-time representation with a sequential processing [Gao et al., 2024, Nie et al., 2023, Woo et al., 2024, Ansari et al., 2024, Dong et al., 2024, Garza and Mergenthaler-Canseco, 2023, Das et al., 2024, Liu et al., 2024, Kuvshinova et al., 2024] models for traditional time series modeling. Adaptations to the baseline transformer includes structuring observations into text with finetuning [Zhang et al., 2023, Zhou et al., 2023], without finetuning [Xue and Salim, 2024, Gruver et al., 2023], or using autoregressive model vision transformers to model unevenly spaced time series data by converting time series into images [Li et al., 2023].

Continuous-time systems are of great interest for learning causal representations using assumptions by using observations to directly modify the system state [De Brouwer et al., 2022, Jia and Benson, 2019]. Variations include intervention modeling with separate ODEs for interventions and outcome processes [Gwak et al., 2020], using liquid time-constant networks [Hasani et al., 2021, Vorbach et al., 2021], or modeling treatment effects with either one [Bellot and van der Schaar, 2021] or multiple interventions [Seedat et al., 2022]. The importance of accounting for external interventions is a particular challenge in clinical data, where external interventions (change in environment due to treatment decisions or clinical context such as ED or ICU) are common in clinical data trajectories.

## 6   Conclusion

In this work we present Trajectory Flow Matching, a simulation-free training algorithm for neural differential equation models. We show when trajectory flow matching is valid theoretically, then demonstrate its usefulness empirically in a clinical setting. The ability to model the underlying

continuous physiologic processes during critical illness using irregular, sparsely sampled, and noisy data has the potential for broad impacts in care settings such as the emergency department or ICU. These models could be used to improve clinical decision making, inform monitoring strategies, and optimize resource allocation by identifying which patients are likely to deteriorate or recover. These use cases will require thorough prospective validation and calibration for specific clinical outcomes, for example using the likelihood of a patient crossing a specific heart rate or blood pressure threshold for decisions on level of care (ICU versus inpatient floors) or specific interventions such as transfusions. In these applications, it will be important to assess and control for bias that may be present due to which patient subpopulations are present in training data.

**Limitations**  Limitations of the method includes the selective utility of integrating memory in clinical settings with high measurement frequency and no current capacity for estimating causal representations, though this will be an important future research direction. Potential harms include the following: erroneous predictions that either results in delayed care or overutilization of the health system. Accurate trajectory predictions have the potential to inform clinical decision-making regarding the appropriate level of care, leading to more timely and appropriate interventions.

**Future work**  We hope to extend our method to cover other types of time series that have periodicity in the components, potentially incorporating Fourier transform [Li et al., 2021] and Physics-Inspired Neural Networks (PINN). Since interpretability is an important factor for clinical reliability, we are developing methods to further elucidate key components affecting the prediction. As well, we hope to incorporate functional flow matching for fully continuous setting [Kerrigan et al., 2024].

## 7  Broader Impact

Our work extends flow matching into the domain of time series modeling, demonstrating a specific instance of clinical time series prediction. In contrast to the large transformer-based models, our method has fewer in parameters and less training time needed. Notably, it scales well with parameters. As well, our parameterization on Stochastic Differential Equations (SDE) allow faster training time than traditional SDE integration.

Accurate timeseries modeling in healthcare has the potential for significant benefits, but also introduces risks. Benefits that could be derived from more accurate prediction of clinical courses include improved treatment decisions, resource allocation, as well as more informative discussions of prognosis with patients or family members. Risks may come from inaccuracies in predictions which could lead to harms by biasing decision making of clinical teams. In the general case of false negative prediction (prediction of trajectories with falsely favorable outcomes) this may lead to undertreatment and in the case of false positive prediction (prediction of trajectories with incorrect detrimental outcomes) or overtreating patients. These inaccuracies may also propagate biases in training data.

To move towards broad impact in the clinical domain, this work will require validation and bias estimates. Furthermore, models deployed in domains with high-stakes prediction require interpretability, which can help identify biases, miscalibration, discordance with domain knowledge, as well as build trust with teams using predictions from the model. At this time, flow-based methods have limited tools for interpretability, and we recognize this as a gap in need of future work.

## Acknowledgements

The authors would like to thank Mathieu Blanchette for useful comments on early versions of this manuscript. We are also grateful to the anonymous reviewers for suggesting numerous improvements.

The authors acknowledge funding from the National Institutes of Health, UNIQUE, CIFAR, NSERC, Intel, and Samsung. The research was enabled in part by computational resources provided by the Digital Research Alliance of Canada (`https://alliancecan.ca`), Mila (`https://mila.quebec`), Yale School of Medicine and NVIDIA.

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

## A Proof of theorems

We first prove a Lemma which shows TFM learns valid flows between distributions with the target prediction reparameterization trick.

**Lemma A.1.** *If $p_t(x) > 0$, $\delta_{data}$ is Lipschitz continuous for all $x \in \mathbb{R}^d$ and $t \in [0,1]$, $\mathcal{L}_{FM}$ and $\mathcal{L}_{TFM}$ are equal,*

$$\nabla_\theta \mathcal{L}_{FM}(\theta) = \nabla_\theta \mathcal{L}_{TFM}(\theta)$$

*Proof.* This proof is a simple extension of Lipman et al. [2023], Tong et al. [2024] which proved $\mathcal{L}_{CFM}$ and $\mathcal{L}_{FM}$ are equal under similar constraint.

Given $\delta_{data} = t_1 - t_0$, we have $u_t(x) = \frac{x_1 - x_0}{\delta_{data}}$ where $t_0$ is the previous time in the time series, and $t_1$ is the current time for inference. For the time series data, we are assuming it to be Lipschitz continuous there exist $L \geq 0$ such that for all $x, y \in \mathbb{R}^n$, $|f(x) - f(y)| \leq L\|x - y\|$.

$$\nabla_\theta \mathbb{E}_{p_t(x)} \|v_\theta(t,x) - u_t(x)\|^2 = \mathbb{E}_{t,q(z),p_t(x|z)} \frac{1}{(1-t)^2} \|\hat{x}_\theta^1(t,x) - x_1\|^2 \tag{18}$$

$$= \nabla_\theta \mathbb{E}_{t,q(z),p_t(x|z)} \frac{1}{(1-t)^2} \left( \|\hat{x}_\theta^1(t,x)\|^2 - 2\left\langle \hat{x}_\theta^1(t,x), x_1 \right\rangle + x_1^2 \right) \tag{19}$$

$$= \nabla_\theta \mathbb{E}_{t,q(z),p_t(x|z)} \left( \frac{1}{(1-t)^2} \|\hat{x}_\theta^1(t,x)\|^2 - 2\left\langle \hat{x}_\theta^1(t,x), x_1 \right\rangle \right) \tag{20}$$

By bilinearity of the 2-norm and since $x_1$ is independent of $\theta$. Next,

$$\mathbb{E}_{p_t(x)} \frac{1}{(1-t)^2} \|\hat{x}_\theta^1(t,x)\|^2 = \int \|\hat{x}_\theta^1(t,x)\|^2 p_t(x) dx$$

$$= \iint \|\hat{x}_\theta^1(t,x)\|^2 p_t(x|z) q(z) dz dx$$

$$= \mathbb{E}_{q(z),p_t(x|z)} \|\hat{x}_\theta^1(t,x)\|^2$$

Finally,

$$\mathbb{E}_{p_t(x)} \left\langle \hat{x}_\theta^1(t,x), x_1 \right\rangle = \int \left\langle \hat{x}_\theta^1(t,x), \frac{\int x_1 p_t(x|z) q(z) dz}{p_t(x)} \right\rangle p_t(x) dx$$

$$= \int \left\langle \hat{x}_\theta^1(t,x), \int x_1 p_t(x|z) q(z) dz \right\rangle dx$$

$$= \iint \left\langle \hat{x}_\theta^1(t,x), x_1 \right\rangle p_t(x|z) q(z) dz dx$$

$$= \mathbb{E}_{q(z),p_t(x|z)} \left\langle \hat{x}_\theta^1(t,x), x_1 \right\rangle$$

Where we first substitute then change the order of integration for the final equality. Since at all times $t$ the gradients of $\mathcal{L}_{FM}$ and $\mathcal{L}_{TFM}$ are equal, $\nabla_\theta \mathcal{L}_{FM}(\theta) = \nabla_\theta \mathcal{L}_{TFM}$

by substitution.

$$\mathcal{L} \mathbb{E}_{t,q(z),p_t(x|z)} \|v_\theta(t,x) - u_t(x|z)\|^2 = \mathbb{E}_{t,q(z),p_t(x|z)} \frac{1}{(\lceil t \rceil - t)^2} \|\hat{x}_\theta^{\lceil t \rceil}(t,x) - x^{\lceil t \rceil}\|^2 \tag{21}$$

$$\mathbb{E}_{t,q(z),p_t(x|z)} \|v_\theta(t,x) - u_t(x|z)\|^2 = \mathbb{E}_{t,q(z),p_t(x|z)} \left\| \frac{\hat{x}_\theta^{\lceil t \rceil}(t,x) - x}{\lceil t \rceil - t} - \frac{x^{\lceil t \rceil} - x}{\lceil t \rceil - t} \right\|^2 \tag{22}$$

$$= \mathbb{E}_{t,q(z),p_t(x|z)} \frac{1}{(\lceil t \rceil - t)^2} \|\hat{x}_\theta^{\lceil t \rceil}(t,x) - x^{\lceil t \rceil}\|^2 \tag{23}$$

$\square$

**Lemma 3.1** *The SDE $dx_t = u_t(x|z)dt + \sigma^2 dW_t$ where $u_t$ is defined in eq. 9 generates $p_t(x|z)$ in eq. 8 with initial condition $p_0 := \delta_{x_1}$ where $\delta$ is the Dirac delta function.*

*Proof.* For simplicity of notation we first show the case where $\lceil t \rceil = 1$.

$$dx_t = u_t(x|z)dt + \sigma^2 dW_t = \frac{1 - x_t}{1 - t}dt + \sigma^2 dW_t \tag{24}$$

which is equivalent to the $d$ dimensional Brownian bridge which has marginal

$$\mathcal{N}((1 - t)x_0 + tx_1, \sigma^2 t(1 - t)) \tag{25}$$

which completes the proof for $\lceil t \rceil = 1$. $\qquad\square$

**Proposition 3.2 (Coupling Preservation)** *Under mild regulatory criteria on $u_t(\cdot|z)$, $p_t$, and $q$, if*

$$\mathbb{E}_{t\sim\mathcal{U}(0,T),z\sim q(z),c\sim q(c|z),x_t\sim p_t(x_t|z)}\|u_t(x_t|z,c) - u_t(x_t|c)\|_2^2 = 0$$

*and $z, q(z), p_t(x|z)$, and $u_t(x|z)$ are as defined in eqs. 6-9 then $\Pi(u)^\star = \Pi^\star(x_{1:T})$.*

*Proof.* We prove the deterministic case with $T = 1$. The extensions to stochastic and $T > 1$ are evident. The couplings are equal if the marginal vector field $u_t(x_t|c) = u_t(x_t|z, c)$ everywhere as the coupling is governed by the push forward flows $\phi(x_0, c) = \int_0^1 u_t(x_t|c)dt$, and $\phi(x_0, c, z) = \int_0^1 (u_t(x_t|z, c)$. If

$$\mathbb{E}_{t\sim\mathcal{U}(0,T),z\sim q(z),c\sim q(c|z),x_t\sim p_t(x_t|z)}\|u_t(x_t|z,c) - u_t(x_t|c)\|_2^2 = 0$$

then $\phi(x_0, c, z) = \phi(x_0, c)$ for all $x_0$ and therefore the couplings of the optimal map are equivalent. We note that this requires exchange of integrals under the same conditions as **Lemma A.1**. $\qquad\square$

Next we show how **(A1)-(A3)** satisfy Prop. 3.2.

- **(A1)** When $c = x_0$ and there exists $T : \mathcal{X} \to \mathcal{X}$ such that $T(x_0) = x_1$ if and only if $\Pi^\star(x_0, x_1)$. We note that this is equivalent to asserting the existence of a Monge map $T^\star$ for the coupling $\Pi^\star$.

  In the two timepoint case, $c = x_0$ is sufficient as long as there aren't two trajectories that have the same $x_0$ but different $x_1$s. Conditioning on this way ensures the conditions of of Prop. 3.2 as the uniqueness property ensures the uniqueness of $u_t(x_t|c)$.

- **(A2)** There exist no two trajectories $x^i$, $x^j$ such that $x_t^i = x_t^j$ for $h + 1$ consecutive observations. In this case notice that this is simply a multi-timepoint extension of **A1** to $c = x_{t-h-1:t-1}$, i.e. conditioned on a history of length $h$. If this is the case then the same reasoning as **A1** applies.

- **(A3)** Trajectories are associated with unique conditional vectors $c$ independent of $t$.
  This satisfies Prop 3.2 by definition.

**Proposition 3.3** *There exists a scaling function $c(t) : \mathbb{R}_+ \to \mathbb{R}$ such that $\mathcal{L}_{\text{target}}(\theta) = c(t)\mathcal{L}_{\text{match}}(\theta)$.*

*Proof.* We start with the matching loss.

$$\mathbb{E}_{t,q(z),p_t(x|z)}\|v_\theta(t, x) - u_t(x|z)\|^2 = \mathbb{E}_{t,q(z),p_t(x|z)} \frac{1}{(\lceil t \rceil - t)^2}\|\hat{x}_\theta^{\lceil t \rceil}(t, x) - x^{\lceil t \rceil}\|^2 \tag{26}$$

by substitution,

$$\mathbb{E}_{t,q(z),p_t(x|z)}\|v_\theta(t, x) - u_t(x|z)\|^2 = \mathbb{E}_{t,q(z),p_t(x|z)} \left\|\frac{\hat{x}_\theta^{\lceil t \rceil}(t, x) - x}{\lceil t \rceil - t} - \frac{x^{\lceil t \rceil} - x}{\lceil t \rceil - t}\right\|^2 \tag{27}$$

$$= \mathbb{E}_{t,q(z),p_t(x|z)} \frac{1}{(\lceil t \rceil - t)^2}\|\hat{x}_\theta^{\lceil t \rceil}(t, x) - x^{\lceil t \rceil}\|^2 \tag{28}$$

completing the proof. $\qquad\square$

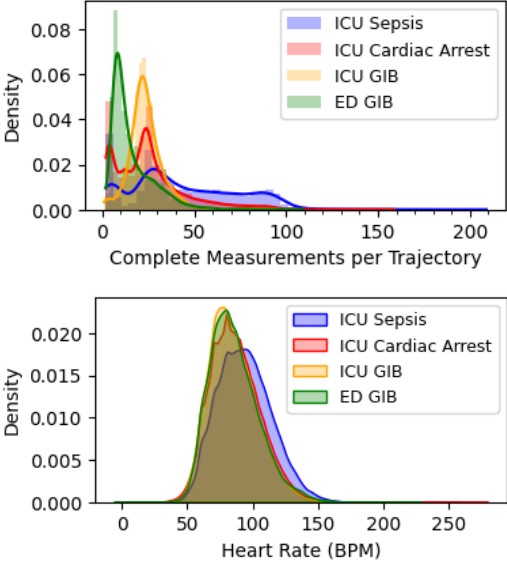

Figure 4: **Left**: Distribution of number of complete vital measurements per patient trajectory within the first 24 hours of admission in each clinical dataset. **Right**: Distribution of raw heart rate values in each clinical dataset.

## B   Experimental Details

### B.1   1D Oscillators

The three oscillation trajectories correspond to $c = 0.25$ (the red trajectory in Figure 2), $c = 2$ (blue), and $c = 3.75$ (green). Before used as an input, $t$ was scaled to between 0 and 1 by dividing by 10.

### B.2   Clinical Datasets

#### B.2.1   Clinical Data Characteristics

In order to accurately model the perturbations in the physiologic signals (mean arterial pressure and heart rate) of the underlying patient states, we need to learn beyond the general trend of the data.

While the physiologic measurements themselves reflect patient status and drive clinical decision making, the degree of variation holds information that goes beyond the snapshot at a single time point. Our approach models the data distribution and stochasticity rather than just fitting the average trajectory. Other time-varying such as treatment conditions and non-time-varying covariates such as underlying disease states may also hold information that may impact the underlying state generating the physiologic signals. Our approach also incorporates this information to inform the trajectory modeling.

The data distribution in the ICU datasets reflect its status as the most resource-intensive clinical setting with increased measurement frequency and data distribution shift towards more abnormal physiologic values (Figure 4). The ED dataset reflects its status as the clinical setting focused on triaging patients, with sparser and physiologic measurements that fall in the normal range.

#### B.2.2   Clinical Data Preprocessing

For each clinical dataset, we modeled patient trajectories formed with heart rate and blood pressure measurements during the first 24 hours following admission. The timeline for each trajectory, originally in minutes, was scaled to a range between 0 and 1 by dividing by 1440. Additionally, heart rate and blood pressure values were z-score normalized to standardize the data.

**Intensive Care Unit Sepsis (ICU Sepsis) Dataset**   The eICU Collaborative Research Database v2.0 [Pollard et al., 2019] is a database including deidentified information collected from over 200,000 patients in multiple intensive care units (ICUs) in the United States from 2014 to 2015. The ICU Sepsis Dataset was created by subsetting the eICU Database for 3362 patients with sepsis as the

primary admission diagnosis (2689 patients in training set, 336 in validation set, and 337 in test set). The following data fields were extracted: patient sex, age, heart rate, mean arterial pressure, norepinephrine dose and infusion rate, and a validated ICU score (APACHE-IV). Each patient's complete pair measurements of heart rate and mean arterial pressure over time form one trajectory to be modeled.

Norepinephrine infusion rates were calculated by converting drug doses or infusion rates to $\mu$g/kg/min, and where drug doses were not explicitly available, the dose was inferred from the free text given in the drug name. Start and end times for norepinephrine infusion were calculated by dividing the dose by the infusion rate. Where there appeared to be multiple infusions at the same time, the maximum infusion rate was taken as the infusion rate. As a conditional input to the models, the norepinephrine infusion doses are then scaled to between 0 and 1 by dividing by the maximum norepinephrine value in the dataset.

The APACHE-IV score, a validated critical care risk score, predicts individual patient mortality risk [Zimmerman et al., 2006]. In data preprocessing, we uses logistic regression of the score against binary hospital mortality data to generate a probability for each patient, serving as an additional input condition for models.

**Intensive Care Unit Cardiac Arrest (ICU Cardiac Arrest) Dataset** This dataset was extracted from the to eICU Collaborative Research Database v2.0 [Pollard et al., 2019] described above to reflect ICU patients at risk for cardiac arrest. This dataset excludes patients who presented with myocardial infarction (MI) and includes variables used in the Cardiac Arrest Risk Triage (CART) score [Churpek et al., 2012]: respiratory rate, heart rate, diastolic blood pressure, and age at the time of ICU admission. As an input to the model, the age was z-score normalized. 51671 patients were included in the training set, with 6459 patients each in the validation and test sets.

**Intensive Care Unit Acute Gastrointestinal Bleeding (ICU GIB) Dataset** The Medical Information Mart for Intensive Care III (MIMIC-III) critical care database contains data for over 40,000 patients in the Beth Israel Deaconess Medical Center from 2001 to 2012 requiring an ICU stay [Johnson et al., 2016]. We selected a cohort of 2602 ICU patients with the primary diagnosis of gastrointestinal bleeding to form the ICU GIB dataset, split into a training set of 2082 patients, and a validation set and a test set of 260 patients each. We extracted the following variables: age, sex, heart rate, systolic blood pressure, diastolic blood pressure, usage of vasopressor, usage of blood product, usage of packed red blood cells, and liver disease. Since the vasopressor and blood product usage are encoded as a binary value and may not represent actual infusion amount that are most likely decaying, we experimented with adding a Gaussian decay to them to use as conditional inputs. Likewise, trajectories to model consist of complete pairs of heart rate and mean arterial pressure (calculated from systolic blood pressure and diastolic blood pressure) measurements.

**Emergency Department Acute Gastrointestinal Bleeding (ED GIB) Dataset** This dataset reflects 3348 patients presenting with signs and symptoms of acute gastrointestinal bleeding to two hospital campuses in Yale New Haven Hospital between 2014 and 2018. The patients were split into a training set, a validation set, and a test set of 2636, 352, and 360 patients. Variables extracted include patient sex, age, heart rate, mean arterial pressure, initial measurements of 24 lab tests, and 17 pre-existing medical conditions as determined by ICD-10 codes. Like ICU Sepsis data, the trajectoires consist of complete pairs of heart rate and mean arterial pressure measurements.

Age, initial lab test measurements (three labs omitted due to missing data), and pre-existing medical conditions were used to train an XGBoost model [Chen and Guestrin, 2016] to predict the binary outcome variable indicating the need for hospital-based care. The resulting probabilities of requiring hospital-based care (outcome of 1) for each patient were then calculated using the trained model and used as conditional input to conditional models in experiments on this dataset.

Of note, the outcome variable was defined as 1 if a patient (1) requires red blood cell transfusion, (2) requires urgent intervention (endoscopic, interventional radiologic, or surgical) to stop bleeding or (3) all-cause 30-day mortality. Labs and medical conditions included in this dataset are listed below. Labs in bold were excluded from the XGBoost risk score calculation due to missing data.

- **Labs:** Sodium, Potassium, Chloride, Carbon Dioxide, Blood Urea Nitrogen, Creatinine, International Normalized Ratio, **Partial Thromboplastin Time**, White Blood Cell Count, Hemoglobin, Platelet Count, Hematocrit, Mean Corpuscular Volume, Mean Corpuscular Hemoglobin, Mean Corpuscular Hemoglobin Concentration, Red Cell Distribution Width,

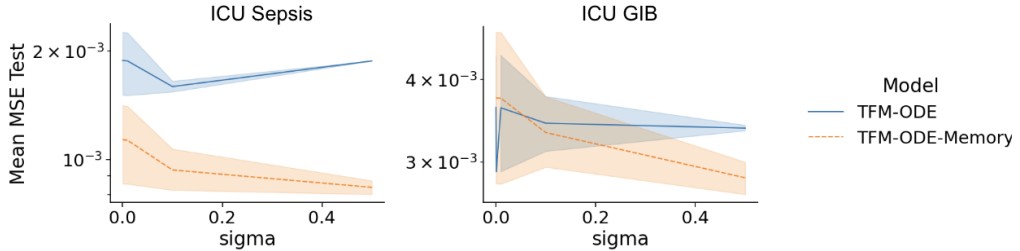

Figure 5: Sigma mean MSE comparison

Red Blood Cell Count, Aspartate Aminotransferase, Alanine Aminotransferase, Alkaline Phosphatase, Total Bilirubin, **Direct Bilirubin**, Albumin, **Lactate**.

- **Previous Medical Histories:** Charlson Comorbidity Index, Cerebrovascular Accident, Deep Vein Thrombosis, Pulmonary Embolism, Atrial Fibrillation, Upper Gastrointestinal Bleeding, Lower Gastrointestinal Bleeding, Unspecified Gastrointestinal Bleeding, Peptic Ulcer Disease, Helicobacter Pylori Infection, Coronary Artery Disease, Heart Failure, Hypertension, Type 2 Diabetes Mellitus, Chronic Kidney Disease, Alcohol Use Disorder, Cirrhosis.

### B.3 Training

#### B.3.1 1D Oscillators

Since the trajectories in this dataset are deterministic and regularly sampled, we deployed only TFM-ODE and applied solely the $\mathcal{L}_{\text{match}}$ loss (i.e, no uncertainty or time predictive loss), as these methods sufficiently address the structured nature of the data to generate proof-of-concept results. The three models presented in Figure 2 all have hidden size of 256, $\sigma$ of 0.1, trained under seed=0 with Adam optimizer with learning rate $1 \times 10^{-3}$ for a maximum of 1000 epochs with early stopping (patience=3) monitoring validation loss.

#### B.3.2 Clinical Data

All the models for clinical data experiments are trained with Adam optimizer. A maximum training time and epochs are set to 48 hours and 300, with early stopping (patience=3) monitoring validation loss. All metrics reported were ran with 5 seeds (0,1,2,3,4) to ensure it is reproducible.

**TFM, TFM-ODE, and ablations** The TFM models were trained with learning rate $1 \times 10^{-6}$ and had $\sigma$ of 0.1. The complete models have hidden size of 256 and memory of 3, while ablation study with a hidden size of 64 and/or no memory was performed (Table 4). The noise parameter for the SDE implementation was set to 0.1 for ablations without $\mathcal{L}_{\text{uncertainty}}$. The hyperparameters $\sigma = 0.1$ and memory=3 for full models were selected through experiments with different values of $\sigma$ and memory (Figure 5 and 6).

**FM** The FM baseline models were trained with learning rate $1 \times 10^{-6}$. All models had a hidden size of 64 with $\sigma$ of 0.1.

**Latent Neural ODE** The latent Neural ODE models were trained with a learning rate of $1 \times 10^{-3}$. 100 GRU units were used for the encoder model and the number of latent dimensions was 2.

**Baseline Neural SDE and Neural ODE** Both baseline models were trained with learning rate $1 \times 10^{-5}$ and had a hidden size of 64.

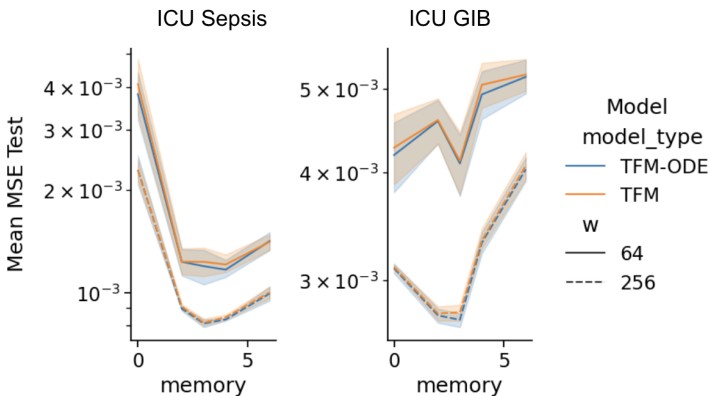

Figure 6: Memory Mean MSE comparison

