# OpenReview forum: "Trajectory Flow Matching with Applications to Clinical Time Series Modelling"
_NeurIPS.cc/2024/Conference — NeurIPS 2024 spotlight_

### Official Review · Reviewer_4EUb · 2024-06-15

**Soundness:** 4
**Presentation:** 3
**Contribution:** 3
**Rating:** 7
**Confidence:** 4

**Summary:**

This paper proposes a trajectory flow matching method for time series data, by using flow matching at each time points. To preserve the coupling of time series, the vector fields are conditioned on history lengths (or even more general $c$). The method also provides ways for model stability, irregularly sampled trajectories and uncertainty prediction. After evaluating the model performance on simple harmonic oscillators, they further apply their method to three clinical time series datasets.

**Strengths:**

The proposed method provides a generative model for time series data, and it takes the benefits of FM, in terms of flexibility, scalability and stability, for time series data modeling.

**Weaknesses:**

The TFM is the model for discrete time series. It would potentially be more useful to model things in a continuous way.

**Questions:**

The paper is clearly written. No major questions.

**Limitations:**

The same as weakness

---

> ### Author Rebuttal · Authors · 2024-08-05
>
> We thank the reviewer for their positive assessment of our paper. We appreciate the reviewer’s insight in the importance of continuous time series modeling. We are also interested in the fully continuous setting and would be interested in adapting ideas from functional flow matching [1] to this domain.
>
> [1] Kerrigan, G., Migliorini, G., & Smyth, P. (2023). Functional Flow Matching (Version 2). arXiv. `https://doi.org/10.48550/ARXIV.2305.17209`

---

> > ### Comment · Reviewer_4EUb · 2024-08-13
> >
> > Thanks for the authors for their response and I would keep my initial rating.

---

### Official Review · Reviewer_s476 · 2024-07-11

**Soundness:** 4
**Presentation:** 4
**Contribution:** 4
**Rating:** 10
**Confidence:** 4

**Summary:**

This paper presents Trajectory Flow Matching, an extension of flow matching to time series. It can model irregular, sparsely sampled, and noisy time series. It trains in a simulation-free manner, bypassing backpropagation through the dynamics. The method is tested on ICU physiological time series, demonstration SoTA performance, and uncertainty prediction.

**Strengths:**

1) A strong research contribution, extending flow matching to time series.
2) Explicit modelling of uncertainty, handles noisy, irregular, and sparsely sampled data.
3) Application to a domain with significant societal benefit.
4) Clearly defined paper, easy to follow, and well-written.

**Weaknesses:**

None of particular note.

**Questions:**

1) It would be interesting to understand if this could be applied to periodic time series like ECG. Do you see ways to introduce this behaviour, and if so, could you call this out as a future research direction?

**Limitations:**

The authors clearly describe the limitations.

---

> ### Author Rebuttal · Authors · 2024-08-04
>
> We sincerely thank the reviewer for their extremely positive response to our work. We are thrilled that you found our TFM to be a strong research contribution. Additionally, we are grateful for your suggestion regarding the future application of our model to periodic time series. We have some preliminary thoughts on this particular behavior that we would like to explore, such as modeling in the Fourier domain, are eager to explore them in future work, and have added this to the future work section of our manuscript.

---

> > ### Comment · Reviewer_s476 · 2024-08-12
> >
> > Thank you for your response, particularly expanding future work and adding the README.md detailed in another review.

---

### Official Review · Reviewer_TMfk · 2024-07-11

**Soundness:** 4
**Presentation:** 3
**Contribution:** 3
**Rating:** 7
**Confidence:** 3

**Summary:**

This paper presents Trajectory Flow Matching (TFM), a simulation-free training algorithm for neural differential equation models. This enables modeling of continuous physiologic processes using irregular, sparsely sampled, and noisy data, all with better scalability.

The authors provide theoretical proof that matching techniques can allow for simulation-free training, and empirical proof of the effectiveness of TFM in clinical settings. An ablation study is also provided to validate components of TFM.

**Strengths:**

- Originality : Bridging the gap between flow matching and differential equation-based models is a novel idea that is carried out rigorously by the authors.
- Quality : Proofs are provided for the theoretical results; experiments are performed on three different real-world datasets; uncertainty prediction is provided by design and helps clinical usability. Responses to the NeurIPS checklist are also extensive.
- Clarity : The paper is easy to parse, as all ideas follow each other logically.
- Significance : SDEs are notoriously expensive to train and this work promises to solve this issue while offering state-of-the-art performance, by a notable margin.

**Weaknesses:**

- The motivation for the paper is that SDEs are not scalable. While this is a known fact, the paper lacks a proper discussion of the impact of simulation-free training on the scalability of TFM.
- The provided code lacks instructions to reproduce the results (empty README).
- Explanability studies would have been welcome to help clinical usability.

**Questions:**

- Did you compare TFM-ODE with other models that are able to process irregular clinical time series?
- What is your reasoning behind modeling specifically the heart rate + MAP pair ?

**Limitations:**

The authors have adequately addressed the technical limitations (memory not always suited depending on the dataset, no causal representations estimations) as well as societal impacts (false predictions).

---

> ### Author Rebuttal · Authors · 2024-08-05
>
> We thank the reviewer for their valuable feedback that helps us further improve our paper. We have added a README, and impact statement on simulation-free training scalability in our manuscript.
>
> We agree having explainability is important for others who may apply our method. We thank the reviewer for bringing this point up as it is quite important in high-stake applications such as healthcare. As such, we have included this in our future work section.
>
> We would like to address some of the reviewers questions below:
>
> > Did you compare TFM-ODE with other models that are able to process irregular clinical time series?
>
> Yes, we compared TFM-ODE’s performance to NeuralODE [1], FM baseline ODE [2], and LatentODE RNN [3], as well as SDE-based models like TFM and NeuralSDE [4]. All these models are able to process irregularly sampled time series. (See Table 1)
>
> > What is your reasoning behind modeling specifically the heart rate + MAP pair ?
>
> We chose the heart rate + MAP pair since they are the most obtainable and non-invasive metric from a patient that provide a clinically relevant representation of patient state trajectories in the management of clinical conditions where hemodynamic monitoring is important (e.g., sepsis and GI bleeding). We chose MAP since it reflects both cardiac output and peripheral vascular resistance; this is used for decisions regarding resuscitation and vasopressor treatment, which are indications for hospital-based treatment. Heart rate may be affected by either cardiac arrhythmias or clinical conditions related to but independent from cardiac output and peripheral vascular resistance. Therefore we used a combination of the two to reflect clinical trajectories.
>
> We hope we have addressed the questions of the reviewer and we would like to thank the reviewer once more for helping us improve the clarity of our responses and propose future directions.
>
> [1] Chen, R. T. Q., Rubanova, Y., Bettencourt, J., & Duvenaud, D. (2018). Neural Ordinary Differential Equations (Version 5). arXiv. `https://doi.org/10.48550/ARXIV.1806.07366`
>
> [2] Lipman, Y., Chen, R. T. Q., Ben-Hamu, H., Nickel, M., & Le, M. (2022). Flow Matching for Generative Modeling (Version 2). arXiv. `https://doi.org/10.48550/ARXIV.2210.02747`
>
> [3] Rubanova, Y., Chen, R. T. Q., & Duvenaud, D. (2019). Latent ODEs for Irregularly-Sampled Time Series (Version 1). arXiv. `https://doi.org/10.48550/ARXIV.1907.03907`
>
> [4] Liu, X., Xiao, T., Si, S., Cao, Q., Kumar, S., & Hsieh, C.-J. (2019). Neural SDE: Stabilizing Neural ODE Networks with Stochastic Noise (Version 1). arXiv. `https://doi.org/10.48550/ARXIV.1906.02355`

---

> > ### Comment · Reviewer_TMfk · 2024-08-08
> >
> > Thank you for your response and actions to improve your paper even further. I appreciate your clarification about the choice of clinical variables.
> >
> >
> > When I mentioned "other models that are able to process irregular clinical time series", I should have been clearer that I had in mind non-differential equations-based models such as STraTS [1],  RAINDROP [2], or Warpformer [3] for example, or even older models such as SAnD [4] or InterpNet [5].
> >
> >
> > Whether you compared your model to these or not, your work still impacts the ODE domain, and my review thus remains positive (Accept), as you have addressed my other concerns.
> >
> > ---
> >
> > [1] S. Tipirneni and C. K. Reddy, “Self-Supervised Transformer for Sparse and Irregularly Sampled Multivariate Clinical Time-Series,” ACM Trans. Knowl. Discov. Data, vol. 16, no. 6, p. 105:1-105:17, Jul. 2022, doi: 10.1145/3516367.
> >
> > [2] X. Zhang, M. Zeman, T. Tsiligkaridis, and M. Zitnik, “Graph-Guided Network for Irregularly Sampled Multivariate Time Series,” presented at the International Conference on Learning Representations, Oct. 2021.
> >
> >  [3] J. Zhang, S. Zheng, W. Cao, J. Bian, and J. Li, “Warpformer: A Multi-scale Modeling Approach for Irregular Clinical Time Series,” in Proceedings of the 29th ACM SIGKDD Conference on Knowledge Discovery and Data Mining, Aug. 2023, pp. 3273–3285. doi: 10.1145/3580305.3599543.
> >
> > [4] H. Song, D. Rajan, J. Thiagarajan, and A. Spanias, “Attend and Diagnose: Clinical Time Series Analysis Using Attention Models,” Proceedings of the AAAI Conference on Artificial Intelligence, vol. 32, no. 1, Art. no. 1, Apr. 2018, doi: 10.1609/aaai.v32i1.11635.
> >
> > [5] S. N. Shukla and B. M. Marlin, “Interpolation-Prediction Networks for Irregularly Sampled Time Series,” ArXiv, Sep. 2019.

---

> > > ### Author Response · Authors · 2024-08-08
> > >
> > > Thank you for your response and for clarifying the methods you had in mind. At this time, we have not compared TFM to these specific methods, and we will look into adding comparisons for the revised version of our paper.

---

### Author Rebuttal · Authors · 2024-08-05

# Global Response

We thank the reviewers for their time evaluating our paper. We are grateful for the thoughtful comments, insights, and potential directions for future work. We have addressed each of the points raised and provided clarifications where necessary. Based on suggestions from the reviewers we have made the following improvements:

* Included README.md with instructions on running the code as well as other code quality improvements. (reviewer TMfk)
* Expanded impact statement of simulation-free training and scalability of TFM to include discussion of scalability relative to simulation-based methods. (reviewer TMfk)
* Expanded discussion of future directions to include modeling periodic time series, functional flow matching for continuous time series, and explainability. (reviewer s476, 4EUb, TMfk)

We look forward to any further feedback and discussions and appreciate the opportunity to improve our work based on your valuable input.

---

### Decision · Program_Chairs · 2024-09-25

**Decision:**

Accept (spotlight)

**Comment:**

The paper presents a significant advancement by bridging flow matching and differential equation-based models, a novel and rigorous approach that is well-supported by theoretical proofs and extensive experiments across multiple real-world datasets. The method’s ability to provide uncertainty prediction and its potential to address the scalability issues of stochastic differential equations (SDEs) while delivering state-of-the-art performance highlight its originality and practical significance. The clarity of the paper, with its logically flowing ideas, and its application to a domain with notable societal benefits further underscore its strengths.

All reviewers are highly supportive of this paper. It stands out by combining substantial algorithmic and theoretical contributions with practical significance.